# Visualizing HIV-1 Capsid and Its Interactions with Antivirals and Host Factors

**DOI:** 10.3390/v13020246

**Published:** 2021-02-04

**Authors:** Morganne Wilbourne, Peijun Zhang

**Affiliations:** 1Magdalen College, University of Oxford, Oxford OX1 4AU, UK; morganne.wilbourne@magd.ox.ac.uk; 2Division of Structural Biology, Wellcome Trust Centre for Human Genetics, University of Oxford, Oxford OX3 7BN, UK; 3Electron Bio-Imaging Centre, Diamond Light Source, Harwell Science and Innovation Campus, Didcot OX11 0DE, UK

**Keywords:** HIV-1, capsid, host proteins, antivirals, small molecules, restriction factors, CryoEM, CryoET, X-ray crystallography, NMR

## Abstract

Understanding of the construction and function of the HIV capsid has advanced considerably in the last decade. This is due in large part to the development of more sophisticated structural techniques, particularly cryo-electron microscopy (cryoEM) and cryo-electron tomography (cryoET). The capsid is known to be a pleomorphic fullerene cone comprised of capsid protein monomers arranged into 200–250 hexamers and 12 pentamers. The latter of these induce high curvature necessary to close the cone at both ends. CryoEM/cryoET, NMR, and X-ray crystallography have collectively described these interactions to atomic or near-atomic resolutions. Further, these techniques have helped to clarify the role the HIV capsid plays in several parts of the viral life cycle, from reverse transcription to nuclear entry and integration into the host chromosome. This includes visualizing the capsid bound to host factors. Multiple proteins have been shown to interact with the capsid. Cyclophilin A, nucleoporins, and CPSF6 promote viral infectivity, while MxB and Trim5α diminish the viral infectivity. Finally, structural insights into the intra- and intermolecular interactions that govern capsid function have enabled development of small molecules, peptides, and truncated proteins to disrupt or stabilize the capsid to inhibit HIV replication. The most promising of these, GS6207, is now in clinical trial.

## 1. Introduction

There are nearly 38 million people worldwide living with human immunodeficiency virus type-1 (HIV), even though HIV is one of the most well-studied viruses and life-saving antiretroviral therapy (ART) has been developed. Understanding the basic mechanisms by which HIV particles are replicated in host cells is immensely important in the context of fundamental research underpinning new antiviral drug development, especially when there is still no early prospect of an HIV vaccine. Current antiretroviral drugs come in eight classes (protease inhibitors, integrase inhibitors, nucleoside reverse transcriptase inhibitors, non-nucleoside reverse transcriptase inhibitors, fusion inhibitors, chemokine receptor antagonists, attachment inhibitors, and post attachment inhibitors) [1,2,3]. These are used in combination in current clinical practice to minimize exerting selection pressure on the viral populations inside patients. Evolving resistance to HIV therapeutics requires that new drugs, and particularly new mechanisms of action, be invented or discovered.

One promising alternative target is the HIV capsid. This is because it is crucial for multiple stages of the viral life cycle, including assembly, maturation, uncoating/disassembly, nuclear import, and integration, thus offering several points of intervention at which an inhibitor could exert its effect. The capsid protects the viral genome from host immune defense; enables trafficking along microtubules; interacts with the nuclear pore to facilitate genomic import into the nucleus; and plays a role in determining sites of viral DNA integration into the host genome [4]. Disrupting one or more of these processes has been shown to affect virus infectivity.

Recent advances in structural biology, including the advent of high-resolution cryo-electron microscopy (cryoEM) and cryo-electron tomography (cryoET), have enabled the detailed visualization of capsid protein (CA) assemblies. Recently published structures illustrate the composition of both the mature and immature multimeric lattice architecture of the HIV capsid [5,6,7,8]. Those same techniques have yielded data showing CA assemblies complexed with host factors and with small inhibitory molecules [9,10,11]. The new studies have ultimately promoted the design and testing of drug candidates.

Nonetheless, there is still a long way to fully understand the contribution of capsid to the viral life cycle. Processes such as “uncoating,” whereby the virus capsid loses its constituents, are not completely understood. Exactly where, when, and how quickly this takes place is still a matter of debate. Some studies suggested that the capsid begins to shed CA in the cytoplasm shortly after fusion [12,13,14,15,16,17], which continues in the nuclear envelope [18,19], while other studies indicated that an intact capsid could dock onto the nuclear pore and uncoat there [20,21,22,23], or that the capsid survives at least partially intact in the nucleus [24]. Despite the differences regarding the initiation of the CA shedding, CA detection in the nucleus has been consistently observed [25,26,27,28]. It has also been reported that reverse transcription spurs uncoating, perhaps by exerting mechanical pressure on the inside of the capsid [29]. As such, there is a great deal of opportunity for future work to determine how the capsid interacts with cellular proteins and how those interactions might be activated or inhibited.

Here we describe recent structural developments in understanding the role of the HIV capsid, in the hope it will highlight gaps in the knowledge for current and future work to fill in.

## 2. Recent Advances Regarding Mature Capsid Structures

The CA protein is a 24 kDa, 231 residue protein with two separately folded components—an *N*-terminal domain (NTD) and a C-terminal domain (CTD), connected by a flexible linker [30]. The former is composed of seven alpha helices and one beta hairpin, while four alpha helices comprise the latter. This has long been established, with high-resolution crystal structures going back to 1996 [31].

CA is the sole constituent of the mature capsid, a conical structure with fullerene-like geometry [6]. There is, however, distinct variation in capsid construction resulting in pleomorphic particles [8]. This is thought to be due to conformational dynamics of the CA protein [8,30,32]. Approximately 1500 CA monomers assemble into 200–250 hexamers and precisely 12 pentamers to make up the mature capsid [6]. In these rosettes, the CTD forms the surface facing the inside of the cone, while the NTD sits atop it facing out toward the environment and arranges into the hexamers and pentamers. It is the NTD-NTD and NTD-CTD interactions which form the hexameric and pentameric rosettes [33,34], while the CTD associates into dimers and trimers to link the lattice together [6,33,34,35,36].

Pentamers induce the requisite curvature to generate the closed ends of the capsule [37]. Additionally, normal mode dynamics simulations show that the pentameric units of the capsid play an important role in capsid stabilization, and those same modeling studies indicate that perturbations in these pentamers result in the initiation of the uncoating process [38]. Seven of the twelve pentamers are predominantly located toward the wider end of the cone, while the remaining five define the narrower end. All-atom molecular dynamics simulations showed chloride ions and sodium ions associated with capsid, at sites of interaction between CA monomers (e.g., the dimerization interface, central pore, and 3-fold symmetry axis) [39], which may indicate involvement in the assembly process, though that has yet to be demonstrated mechanistically. More interestingly, those same simulations showed translocation of ions through the capsid into the internal space, demonstrating possible channels through which nucleotides could transit to facilitate reverse transcription in vivo [39,40].

These insights have come from a variety of structural techniques summarized in Figure 1. Recently, cryoEM and magic angle spinning (MAS) NMR have enabled visualization of wild-type capsid assemblies, but that is a relatively recent phenomenon. Before the resolution revolution which has made cryoEM a viable structural technique, X-ray crystallography provided the first clear description of the CA monomer and its hexameric and pentameric assemblies. Even now, it provides high-resolution insight.

### 2.1. X-ray Crystallography

Interactions between the subunits of CA, in both intra- and intermolecular contexts, are crucial for the formation of both CA hexamers and pentamers, but the understanding of these interactions has evolved significantly over the years. In a relatively early crystal structure of Cys-crosslinked CA hexamers, Met39 and Ala42 comprise elements of a hydrophobic interface that forms part of an 18 helix barrel [33,34]. The analogous barrel in the Cys-crosslinked pentamer crystal was shown to have 15 helices, with largely the same aliphatic contact (Figure 1b) [33]. This arrangement of the pentamer, however, has since been shown to be an artifact of the crosslinking. CryoET subtomogram averaging has subsequently shown a substantially different and more native pentameric structure with 10 helices in the barrel [8].

Specific interactions in the pentamer have also been identified as relevant to capsid assembly. In particular, there is a hydrogen bond between E28 in helix 1 of one NTD and K30′ on helix 1′ of the NTD from the adjacent CA monomer which was found predominantly in pentamers, based on analysis of PDB entries 3J3Q and 3J3Y [41]. The H bond in question could not be identified in relevant crystal structures but this conflict is thought to be an artifact of the flatness of the crystalline lattice.

A more recent crystal structure of full-length wild-type hexameric CA was reported at 2.4 Å, which describes a lattice structure dependent on an adaptable hydration layer (Figure 1a) [30]. These water-mediated hydrogen bonds were observable at the side chains of conserved residues at the 2-fold interface (Glu175, Ser149, Trp184), and backbone carbonyls at the 2-fold (Gln176) or 3-fold (Ile201, Ala204) interface. This is in contrast to cryoEM and NMR studies which have suggested that hexamers interact via hydrophobic interfaces [6,7,32]. This conflict could be due to the flatness of the crystalline lattice, as the more native shape of the tubular assemblies used in cryoEM and NMR, or due to different buffers used for crystallization and cryoEM/NMR.

However, while X-ray crystallography will continue to permit generation of high-resolution structures, cryoEM and cryoET have in recent years begun to overcome the limitations of the harsh conditions imposed by crystallography. Crystal lattices of CA are flat. The viral capsid is conical—a feature not easily recapitulated by crystallography. The conditions used to crystallize CA do not bear close resemblance to the cellular milieu. CryoEM samples, while frozen, are derived from aqueous conditions closer to that of the living cell. For these reasons and more, cryoEM and cryoET have taken on an increasingly important role in determining the interactions of CA lattices.

### 2.2. CryoEM/CryoET

Owing to the apparent benefit that cryoEM offers, it has been an important methodology for structural analysis of large assemblies such as the HIV capsid. Early work by Li et al. from in vitro assembled CA tubes showed a surface lattice of CA hexamers and suggested a fullerene cone model for the capsid architecture [42]. This was followed by a report integrating NMR data of a CTD-CTD dimer with a cryoEM map of tubular CA assemblies at a medium resolution [35]. The NMR structures of the CTD dimer correlated well with the electron density map from the cryoEM data, revealing that the CTD dimer pair are oriented differently than had been reported by previous crystal structures, which were poorly defined in that area [34,35]. The cryoEM/NMR report was also the first to identify the new CTD trimer interface as an integral interaction point between three adjacent hexamers [35]. This trimer was asymmetric and indicated interactions between helix 10 of one CTD and helix 11 of its neighbor CTD. The structural data and subsequent mutagenesis studies determined the interacting resides of helix 10 to be Lys203, Ala204, and Pro207, while the interfacing residues of helix 11 were Glu213, Thr216, Ala217, and Gln219. Finally, the variability in the interdomain and intermolecular interfaces were suggested to account for the curvature in tubular and conical CA assemblies.

Subsequently, the cryoEM structure of a tubular CA assembly was resolved to 8.6 Å, revealing individual α-helices [6]. Building from the new cryoEM data and a 3D structure of a native HIV core by cryoET, an all-atom model of the entire mature capsid was derived through molecular dynamics simulations [6]. This model demonstrated the importance of a three-helix bundle (H10 of the CTD) at the trimer interface. The bundle contains interacting hydrophobic residues (Ile201, Leu202, Ala204, and Leu205) which stabilize the trimer of dimers. These are surrounded by salt bridges that also contribute to this effect (e.g., Lys203 and Glu213). The cryoEM and modeling data also indicate that the incorporation of pentamers induces a tightening of the trimer interface. This results in the increased curvature necessary to close the cone at both ends.

Unlike the native capsid, these in vitro assembled CA tubes contain no pentamers. To understand the organization of native HIV-1 cores, cryoET with subtomogram averaging has been applied to study such pleomorphic assemblies, and structures hexameric and pentameric CA were reconstructed from wild-type cores (Figure 1c) [8]. The subtomogram averaged hexamer at 6.8 Å resolution closely resembled the fully hydrated wild-type and cross-linked hexamer crystal structures [8,34]. By contrast, the pentamers (resolved to 8.8 Å) differed considerably from the cross-linked crystallographic depictions [8,33]. The cryoET pentamer structure clearly showed that the central pore was comprised of 10, not 15 helices, with helix 3 excluded from the interface between monomers due to the 19° rotation of the NTD. The CTD-NTD interface differed from the crystal structure as well, resulting in an altered binding pocket for small molecule inhibitors. However, the electropositive Arg18 ring within the pore was preserved.

While the cryoET was on the native core, a six-fold symmetry was applied during subtomogram averaging to achieve the 6.8 Å resolution. A near-atomic resolution, 3.6 Å structure of a tubular capsid assembly has recently been reported, from which atomic models of the hexameric capsomere were derived [7]. From this, it is evident that the CA hexamer is asymmetric and itself curved at an angle of approximately 20° (Figure 1d–f). The flexibility of the NTD-CTD linker creates this bend and asymmetry. While the previous cryoET structure indicated that tilts and twists between neighboring hexamers could generate capsid curvature, this new structure argues that instead the hexamers themselves are intrinsically curved, and the bend in the capsomere instigates the varying curve of the capsid lattice in wild-type HIV [7,8]. The variation in that curvature can then be attributed to differing angles within individual hexamers, which were observed in CA tubes of different helical symmetry. Through this diversity, a single CA protein can make up a vast number of differently shaped and sized capsids with a continuous curvature.

Further, cryoET has offered insights into the behavior of capsid cores after fusion with the membrane of a host cell rids them of their lipid envelopes [43]. Christensen and colleagues demonstrated that reverse transcription takes place inside an intact or nearly intact capsid, using capsids derived from wild-type virions in an in vitro system [44]. The newly reverse-transcribed DNA bulged out of the capsid as it broke down over the course of 8–10 h. This offered support for the concept that progressive uncoating is driven at least in part by increased pressure due to the completion of reverse transcription [16,29,45].

In the same study, the addition of GS-CA1, which destabilized the capsid integrity, inhibited the synthesis of late transcripts, though the initiation of reverse transcription remained largely intact [44]. By contrast, high levels of inositol hexakisphosphate (IP_6_), which stabilizes the capsid, diminished viral DNA integration. This indicated a hyperstable capsid inhibits integration and suggests some degree of capsid disassembly is required for this process.

CryoEM has implications for the design of therapeutics as well. One report of an engineered CA protein discussed its self-assembly into hexamers containing stabilizing disulfide bonds and then into icosahedral particles with a 40 nm diameter [45]. They contained 12 pentamers and 30 hexamers as determined by cryoEM reconstruction. The modified CA contained the V3 loop from the envelope protein, bound to V3 neutralizing antibodies, and induced immunogenicity in mice. As such this seems to be a possible design avenue for an HIV vaccine.

### 2.3. NMR

NMR experiments allow for examination of the protein dynamics that are thought to contribute to the construction and pleomorphism of the HIV-1 capsid. MAS NMR, in particular, has improved considerably for large protein assemblies in recent years, demonstrated with the utility of both 19F MAS NMR, whereby 19F is substituted into the indole ring of a tryptophan residue in CA, and dynamic nuclear polarization (DNP) MAS NMR, in which signal intensity is increased by the addition of a polarizing agent [46,47,48].

In this vein, MAS NMR demonstrated empirically the flexibility inherent in the CypA binding loop of CA using [15] *N* chemical shift anisotropy tensors [49]. This line of investigation has culminated in the generation of an atomic model of CA hexamer, built from DNP MAS NMR data with constraints from an 8 Å cryoEM map and assisted by molecular dynamics simulations [32]. Consistent with the high-resolution cryoEM structures of tubular assemblies, their model suggested that the curvature of the tube resulted in large part from flexibility in the linker region, with additional conformational variabilities of the dimer and trimer interfaces [7].

Further, other MAS NMR data suggest that the dimer interface may differ between CA assemblies of different geometries [50]. Curved and flat lattices show variation in the signals from amino acid side chains which indicate this to be the case. Signals for the side chains of Trp184 and Met185, both crucial to the dimer interface, were distorted or absent from spectra of the CA sheets, while they were clearly resolved in spectra of tubes and spherical assemblies. This supports the concept that the dimer interaction is governed to some extent by the positioning of the Trp184/Met185 side chains. Alteration in this packing modulates the curvature of the lattice. However, the dimer interface is clearly defined in crystal structures from flat CA assemblies [30], so perhaps the lattice was insufficiently ordered to define the dimer interface in the sheets used for this MAS NMR experiment.

In the same study were also variations in the side chain signals from Trp80 and Trp133 between the differently shaped constructs, indicating that the intramolecular helices exhibit some variation in position between assemblies [50]. However, there are no major conformational changes in CA to which the different lattice structures can be attributed, at least in the solid state. It is thought that disordered regions of the protein may contribute to this phenomenon, but that couldn’t be ascertained by MAS NMR [50,51,52].

## 3. Structures of the Capsid in Complex Host Cell Factors

Numerous virological, biochemical, and structural studies have shown that the HIV capsid regularly interacts with host cell factors. The best characterized binding partners are Cyclophilin A (CypA); Trim5α/TrimCyp; MxB; cleavage and polyadenylation specificity factor 6 (CPSF6); and nucleoporins 153 and 358 (Nup153, Nup358) [7,53,54,55].

### 3.1. CypA

Canonically, the peptidyl-prolyl cis-trans isomerase CypA binds CA via a proline-rich loop that stretches from Pro85 to Pro93, resulting in the incorporation of CypA into the mature viral particle [56]. Abolishing the interaction, either by Cyclosporin A (CsA) or mutation in CA (G89V or P90A), modulates viral infectivity and inhibits replication in a cell-type dependent manner [57,58,59].

MAS NMR has shown that the notable dynamics of the CypA binding loop are markedly attenuated upon binding of CypA [51]. Even more interestingly, the mutants A92E and G94D, which don’t require CypA interaction such as the wild-type, demonstrate similar levels of conformational immobility to the WT CA-CypA complex.

However, cryoEM and all-atom molecular dynamics have suggested a second binding site for CypA at the interface between two hexamers [60]. Most recently, a 4.0 Å cryoEM structure showed definitively that CypA recognizes not just the CA protomer, but 3 different monomers belonging to adjacent hexamers simultaneously (Figure 2d,e) [7]. One monomer has an interface with CypA by the classical CypA binding loop, but the other two interactions are at non-canonical sites. There are also two patterns of binding—one in which the CypA associates with the dimer interface, and one in which one CypA binds above the dimer interface and another CypA binds above the trimer interface. In the CypA binding mode involving the dimer interface, amino acid stretches Glu43-Lys44-Gly45-Phe46 and Pro23-Ile124-Pro125 appear to be particularly important to the non-canonical interaction. This multi-modal binding of CypA to the assembled capsid explains its stabilizing effect on the viral capsid.

### 3.2. Trim5α/TrimCyp

In 2004, two proteins known as Trim5α and TrimCyp were first identified as barriers to HIV infection in rhesus macaques and owl monkeys, respectively [61,62]. Trim5α is composed of four domains—RING (an E3 ubiquitin ligase), B-box 2 (facilitates assembly into multimeric structures), coil-coil (responsible for dimerization), and B30.2 (PRY/SPRY) (binds retroviral capsids) [63]. TrimCyp is very similar, except that the PRY/SPRY domain has been replaced with a CypA domain.

A crystal structure of the rhesus Trim5α PRY/SPRY domain showed it folds into an *N*-terminal α helix alongside two antiparallel β sheets of seven strands each, which come together in a rough β sandwich [64]. Within this structure, there are four variable regions (V1-V4), the first three of which contain the residues which interface with the HIV capsid. V1-V3 together form an exterior surface which contains a flat, triangular space of approximately 40 Å, though the variability in conformation of this region can extend the capsid binding interface to as much as 70 Å.

Structural lines of evidence (X-ray crystallography and cryoEM) indicate that when rhesus Trim5α binds the HIV capsid, it disrupts capsid stability, causing disassembly and initiating the innate immune response [64]. There is analogous Trim5α activity in human cells, though attenuated, and it has been discovered that the reduced human Trim5α effect is due to the presence of CypA [65]. The increased stability conferred by binding of CypA to the CA assembly, or the accessibility of Trim5α to capsid in the presence of CypA, could impede the antiviral activity of Trim5α [65,66].

However, an alternative mechanism has been proposed and supported with biochemical experiments. These indicate that Trim5α operates in a two-stage process. The first step relies solely on binding the capsid and has been shown to inhibit viral infection in its own right, but the second step is dependent on the proteasome [67,68,69]. When Trim5α and TrimCyp bind the capsid, they are destabilized and subsequently degraded [70]. Ubiquitination of Trim5α prevents the build-up of reverse transcripts, and proteasomal inhibition abrogates that activity [69]. The ubiquitination of Trim5α also induced the classic premature disassembly of viral cores, while Trim5α conjugated with a ubiquitinase had no such effect [71].

CryoEM and cryoET in particular have offered insights into the molecular mechanisms by which this immune restriction factor interacts with the capsid [64,72,73]. CryoEM and biochemical data show that monomeric rhesus Trim5α can bind the capsid, but the same isn’t true of human Trim5α [64]. Human Trim5α requires multimerization to efficiently interface with the capsid.

By the same token, monomeric CA has a low affinity for human Trim5α [72]. Trim5α recognizes motifs that appear when CA is organized into assemblies [74]. Furthermore, cryoEM has shown that rhesus Trim5α disrupts the capsid assembly at interfaces between hexameric capsomeres [74]. MAS NMR and MD simulation have shown that Trim5α induces an increase in the rigidity of the capsid when it binds and structural and dynamic changes at the NTD-NTD interface, as well as other key intermolecular interfaces [75].

CryoET studies have shown Trim5α forms a lattice which superimposes itself over the capsid lattice [72]. This increases the avidity of the two structures for one another substantially. The Trim5α lattice is dynamic, a characteristic enabled by constrained diffusion over the capsid, and interfaces with CA in the vicinity of the CypA binding loop. CryoET has also shown lattice formation to be a hierarchical process [73]. A few Trim5α monomers assemble together as a starting point from which higher-order structures can develop.

### 3.3. MxB

The MxB protein is a dynamin-like restriction factor which inhibits HIV proliferation by binding to the capsid [76]. Its activity is induced by type 1 interferons (particularly α-interferon), which are known to be involved in HIV defense [77]. MxB is comprised of 3 domains (a stalk responsible for oligomerization; a GTPase; and a bundling signaling element domain), but it is the short *N* terminal tail which facilitates binding to the HIV capsid [78,79].

Modeling and mutagenic studies have shown that oligomerization is compulsory in order for MxB to efficiently bind the capsid assembly [80]. A 4.6 Å cryoEM structure of the full-length MxB, in a fusion construct with Mannose Binding Protein (MBP) has also been reported, in which it was evident that MxB polymerizes into a right-handed helix when GDP bound [81]. Binding of GTP caused the helix to dissociate until the GTPase domain hydrolyzed it to GDP, at which point the helix reformed. This oligomerization, in addition to the *N*-terminal region, is necessary for MxB restriction of the HIV capsid, but not the L4 loop and GTPase which are crucial for activity of the related protein MxA against other viruses such as influenzas [80,82,83]. Other studies showed deletion of the C-terminal 100 amino acids or a substitution L661K on the MxB protein inhibited its ability to assemble and also diminished its propensity to attenuate HIV-1 infection [80,84].

There are four interfaces by which MxB self-associates [81]. Interface 2 is a highly conserved dimeric interface analogous to that of MxA and truncated MxB crystal structures [81,85]. Interfaces 1 and 3 facilitate oligomerization [81]. Interface 1 is comprised of a contact between the stalk of one dimer and the bundling signal domain of the adjacent dimer, mediated largely by hydrophobic interactions and a nearby salt bridge. Interface 3 is between the C terminus of stalk helix 1 and the L1 and L2 loops. The interactions which delineate the interface are also largely hydrophobic, as in Interface 1. Interestingly, mutations disrupting Interface 3 inhibited both oligomerization of MxB and its HIV restriction activity. Interface 4 mediates higher order helical assembly and is composed of contact between the GTPase of one dimer and the stalk-hinge region of a dimer 5 units down.

Additionally, it has been reported that CA protomers, and even hexamers, are insufficient to bind MxB [78]. Instead, molecular modeling and biochemical studies indicate that MxB binds at the junction between three adjacent hexamers, in a negatively charged pocket defined by 12 glutamate residues (3 copies each of Glu71, Glu75, Glu212, and Glu213) [86]. It is thought that the Arg11-Arg12-Arg13 sequence on MxB binds into this negative inter-hexamer well, thus creating an electrostatic attraction between the two proteins [86].

### 3.4. Nup358/Nup153

Lentiviruses such as HIV-1 are notable for their ability to infect cells which are not dividing. They do this by entering the nucleus, and as such, it is not surprising that HIV-1 has been shown to associate with nuclear pore proteins [87].

HIV-1 has been shown to interact with nucleoporin 358 (Nup358), a nuclear pore protein containing a CypA homology domain with isomerase capabilities [87]. Nup358 is a transport channel which forms part of a larger complex that associates with the cytosolic ring of the nuclear pore, as shown by cryoET [88]. Crystal structures of its *N* terminal domain and C terminal domain have been determined separately, at 0.95 Å and 1.75 Å, respectively [89,90]. The NTD is comprised of alpha helices, within which are three tetratricopeptide repeats [90]. The repeats are bound on both sides by an amphipathic helix. They also appear to lack the traditional peptide-binding groove due to an extended conformation. In addition, the NTD surface has a largely positive electrostatic potential and has been shown to bind RNA. The CTD resembles a cyclophilin, though the arrangement of the active site is unusual and insensitive to inhibition with cyclosporine [89]. It is this active site which binds CA, albeit weakly, via the CypA binding loop [87,89]. This supports the hypothesis that Nup358 facilitates the transfer of the HIV core into the nucleus, which has also been shown by fluorescence microscopy [21].

In addition, studies showed HIV-1 infection is dependent on Nucleoporin 153 (Nup153) binding [11,91], which has been subsequently linked to nuclear import [79,80]. There have been reports of crystal structures of hexameric CA co-crystallized with a short peptide based on Nup153 (comprised of residues 1407–1429) (Figure 2c) [9,90]. It is a stretch of phenylalanine-glycine repeats which interact with a CA hexamer, lodged in the cleft between an NTD on one monomer and a CTD on another [9]. This NTD-CTD cleft, involving highly conserved Pro34, Ile37, Pro38, Asn53, Leu56, Asn57, Val59, Val142, and Gln176, interacts with Nup153 [92]. The same binding pocket is also used to recruit the nuclear targeting cofactor CPSF6 and is competed for by the antiretroviral compounds PF74 (Figure 2a–c) [9].

### 3.5. CPSF6

Cleavage and polyadenylation specificity factor 6 (CPSF6) interacts with CA and is required for nuclear entry of the viral particle [54]. In CPSF6 depleted macrophages, the viral genome, still associated with a partially intact capsid, lodged inside the nuclear pore rather than exiting into the nuclear compartment [54]. CPSF6 has also been shown to facilitate trafficking of the HIV to nuclear speckles, the location where retrotranscribed viral DNA preferentially integrates into the genome [93]. When the capsid fails to interact with CPSF6, viral DNA integration is dysregulated. Without CPSF6, the virus localizes to the nuclear periphery and integrates into transcriptionally inactive heterochromatin, instead of trafficking to the middle of the nucleus and integrating into euchromatin as it should [94].

Multiple crystal structures of a CA hexamer bound to a CPSF6 peptide fragment were determined, showing that it binds to the same NTD-CTD cleft between protomers as Nup153 (Figure 2c) [9,95,96]. Nup153 and CPSF6 share the requisite phenylalanine-glycine repeat sequence which binds that interface in the hexamer, and as a result, have the same affinity for that region (though the CA amino acids with which they interact overlap but are not identical). Given the relative locations of Nup153 and CPSF6, this suggests that CPSF6 replaces Nup153 in binding the capsid to once Nup153 has facilitated its transit into the nucleoplasm.

Truncation of the C terminal nuclear targeting sequence of CPSF6 (a domain rich in Arg and Ser), results in localization of the abbreviated protein to the cytoplasm [97]. Intriguingly, this construct inhibits HIV infectivity by binding the capsid and sequestering it away from the nucleus. It was also shown by transmission electron microscopy (TEM) that CPSF6-358, as the truncated protein is termed, binds CA tubes as an oligomer and disrupts the assembly into small fragments [97]. This, combined with findings from live and fixed cell imaging, indicates that oligomeric CPSF6-358 prompts cytosolic capsid permeabilization and premature uncoating.

## 4. Structural Insights into the Effects of Small Molecules

Based on the accumulated evidence suggesting that viral replication and infectivity are dependent on the appropriate function and assembly of the capsid, there has been a flurry of research investigating the effect of small molecules on this process. Inhibitors include those targeted at the CA-SP1 junction in the immature particle (termed maturation inhibitors), but also small molecules which bind the NTD-CTD interface within the capsomere of the mature viral particle [9,98,99]. Further, there is a host-derived small molecule, inositol hexakisphosphate, which has recently been shown to be important for the appropriate construction of new viral particles [100].

### 4.1. Inositol Hexakisphosphate (IP6)

IP6 is a small molecule host factor that has been shown to bind both the immature Gag lattice and the mature capsid, and indeed is packaged into the mature virion at a ratio of approximately 300 monomers per viral particle [100,101]. In doing so IP6 facilitates assembly of the immature Gag hexamer followed by rearrangement into the mature capsid once CA has been liberated from the polyprotein [100,102].

IP6 is present in relatively high concentration in human cells (approx. 40–110 μM) and binds CA in the *myo* position (chair conformation with one axial phosphate and the other five equatorials) [100,103]. Crystal structures and all-atom molecular dynamics indicate IP6 stabilizes the six-helical bundle which determines the hexameric lattice formation of the Gag protein by neutralizing the positive charge of two lysine rings (Lys158 and Lys227 on each Gag protomer). In the absence of IP6, immature HIV cores bind IP5 without reducing infectivity (though substantially fewer virions were produced) [104]. However, Lys227Ile Gag mutants, which cannot bind inositol phosphates at all, generate unstable capsids which do display a comparatively reduced infectivity [104].

IP6 was found to promote assembly of the mature CA protein by neutralizing a similarly positive ring at the center of capsomer, though this time the amino acid in question was Arg18 (Figure 2c) [101]. The crystal structure of the mature hexamer bound to IP6 showed that the polyanion has two interfaces with the capsomere [101,102]. One is above the positively charged ring, in a pocket delineated by the *N*-terminal β-hairpins of each monomer. The other binding site is below the arginine ring. Biochemical experiments show IP6 binds the capsid with somewhere between a 1:1 and a 2:1 ratio, which can be explained by the supposition that hexamers have the capability to bind the polyanion at either or both interfaces described in the crystal structure [105,106]. By contrast, molecular dynamics suggest that the pentamer binds only one molecule of IP6, above the arginine ring [106]. The same modeling also indicates that IP6 preferentially stabilizes pentamers, which have been shown to be important for both the curvature and structural integrity of the capsid. This offers some insight into how CA assembles to form conical fullerenes in vivo, and yet helical tubes in vitro without the pentamers present [102].

This binding then stabilizes the mature capsid after entry to the cell. The increased stability ensures reverse transcription can occur inside undisturbed by the host immune system [101,107]. Without IP6, reverse transcription is attenuated and cellular infection impaired in vivo and in vitro due to the instability of the capsid [85,87]. Substitution of hexacarboxybenzene had a similar effect. Going forward this system will aid in carrying out studies on reverse transcription under less artificial conditions.

### 4.2. PF74 and GS-CA1/GS6207

PF3450074 (a Pfizer compound colloquially termed PF74) was first reported to exhibit antiretroviral activity in 2010, in conjunction with a crystal structure that showed it binds the CA NTD [108]. Subsequently, it was shown to competitively bind to the same site where CPSF6 interfaces with the CA capsomere [96]. A further step in determining the mechanism of action for PF74 came about when new crystal structures of the compound bound to a CA hexamer showed that it occupied a binding pocket between the NTD of one protomer and the CTD of the adjacent one (Figure 2a–c) [9,97]. These same studies confirmed that CPSF6 interfaces with the hexamer at the same location, as does Nup153. Recent studies have shown evidence that PF74 binding excludes viral replication complexes from nuclear speckles, demonstrating the importance of the CA-CPSF6 interaction in targeting viral DNA integration [109].

PF74 has a 50% effectiveness concentration (EC_50_) of 1.239 ± 0.257 μM and a 50% cytotoxicity concentration (CC_50_) of 32.2 ± 9.3 μM [110]. Unfortunately, its low metabolic stability [111,112,113,114,115] makes it unsuitable for further development into a therapeutic, and as such there have been ongoing medicinal chemistry efforts to generate analogues with balanced potency and metabolic stability [111,112,113,114].

One such effort led to the development of GS-CA1. Its reported EC_50_ is 240 ± 40 pM and it has a CC_50_ of greater than 50 μM, making it more potent and less cytotoxic than PF74 [110]. Low solubility and significant metabolic stability suggest it has promise as a long-acting antiretroviral therapy, though poor solubility typically suggests low bioavailability. The length of activity however was demonstrated by injection into a humanized mouse model, where it provided superior results to those of rilpirivine, a long-acting nonnucleoside reverse transcriptase inhibitor. Crystal structures, cryoEM, hydrogen-deuterium exchange experiments, and molecular modeling show that GS-CA1, and its analogue GS-6207, bind to the same pocket where PF74 interfaces with CA (Figure 2c) [10,99]. This means these, too, exclude the binding of CPSF6 and Nup153 from the same region. The GS compounds, and the inter-protomer interactions that they enforce, stabilize the capsid, preventing appropriate uncoating and genome integration [10].

## 5. Outlook

In a phase I clinical trial, GS6207 demonstrated substantial reduction in patient viral load over both the short term (9 days) and medium to long term (6 months) subsequent to a single 450 mg subcutaneous dose [116]. A phase 2/3 trial is currently in progress. Even with this success though, there remains medicinal chemistry work to develop alternative PF74 analogues with better metabolic stability and potency.

There is additional work ongoing to target other sites relevant to capsid assembly. CAI, a 12 amino acid peptide first reported in 2005, binds a hydrophobic groove in the dimer interface to prevent CA assembly [117]. However, peptides are not especially tractable drug candidates due to their lack of penetrance into the cell. Efforts continue to create an analogue with therapeutic potential, but the dimer interface is nonetheless a promising target to inhibit capsid assembly [118,119]. Other drug candidates targeting CA include ebselen, a small molecule which inhibits CA dimerization (thought to covalently cross-link the CTD with a selenyl-sulfide bond with Cys198 and Cys218), and C1, which inhibits mature lattice formation without affecting Gag cleavage [120,121].

Whether GS6207 goes to market or another follows in its footsteps, it seems plausible that the next antiretroviral approved for use in humans will target the HIV capsid.

## Figures and Tables

**Figure 1 viruses-13-00246-f001:**
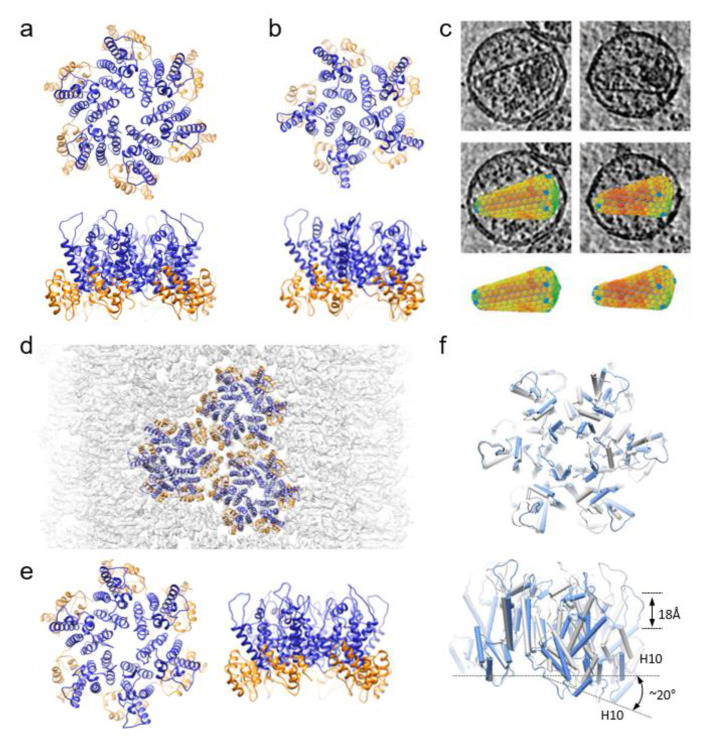
Structures of HIV-1 capsid assembly units. (**a**,**b**) Crystal structures of wild-type CA hexamer (PDB 4XFX) (**a**) and crosslinked CA pentamer (PDB 3P05) (**b**) viewed from top and side. CANTD-blue, CACTD,-gold (**c**) CryoET and subtomogram averaging of hexamers and pentamers from native virions. (**d**,**e**) overlapped with density map (EMDB-10299) (**d**) CryoEM structure of of HIV-1 CA tubular assemblies with CA hexamer structure (PDB 6SKK) (**e**). (**f**) Comparison of curved, asymmetric cryoEM hexamer (PDB 6SKK, blue) with the flat six-fold symmetric crystal hexamer (PDB 4XFX, grey).

**Figure 2 viruses-13-00246-f002:**
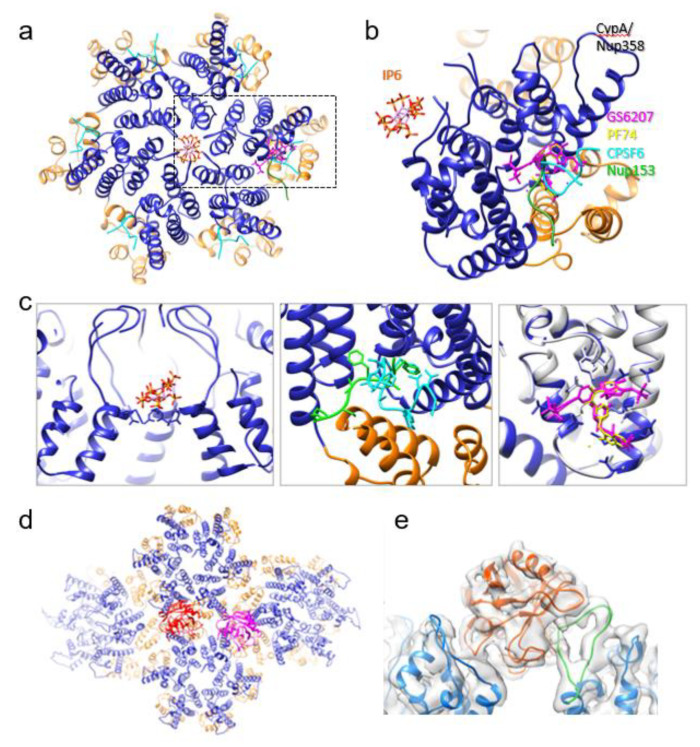
Interactions between HIV-1 capsid and host factors and small molecules. (**a**) Crystal structures of wild-type CA hexamer (PDB 4XFX) superimposed with IP6 (red), CPSF6 peptide (cyan), Nup153 peptide (green), PF74 (yellow) and GS6207 (pink). (**b**) An enlarged view of the boxed region in **a**. (**c**) Detailed interactions between capsid and IP6 (left) CPSF6/Nup153 (middle), PF74/GS6207 (right). (**d**) Interaction between CypA and capsid. Two CypA molecules (red and pink) can simultaneously bind to CA assembly above the dimer and trimer interface, respectively. (**e**) A side view CypA/CA complex, overlaid with the density map, above the dimer interface.

## Data Availability

Not applicable.

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
