# Peer review of "Visualizing HIV-1 Capsid and Its Interactions with Antivirals and Host Factors"

_viruses, 2021, doi:10.3390/v13020246_

Round 1

Reviewer 1 Report

This is a very well written review on an important topic. HIV-1 CA is an emerging target for developing mechanistically novel antivirals. Understanding the structural and molecular basis of CA functions and their inhibition underpins promising drug discovery efforts. The manuscript is well structured, informative and insightful, and its publication will significantly benefit readers of Viruses. A few suggestions for a minor revision:

  • Lines 35-38. “six classes of current HIV-1 drugs”: the authors should consider adding the other two classes, attachment inhibitor (fostemsavir) and post attachment inhibitor (ibalizumab).
  • Line 243. “maner”: change to “manner”.
  • Lines 277-279. “It isn’t, however, altogether clear whether CypA protects HIV from Trim5a, or whether Trim5a disrupts the connection between CypA and the capsid46,47”: this may have misrepresented the cited work. The title of citation #46 reads “Cyclophilin A Protects HIV-1 from Restriction by Human TRIM5a”; and the main conclusion of citation #47 reads “Overall, our results reveal that CypA binding to the core protects HIV-1 from TRIM5αhu restriction” which is consistent with #46. The authors may have misunderstood the disruption of CypA in citation #47—you could disrupt CypA-CA by using cyclosporin A or via mutations, but the effect of the disruption is dependent on Trim5a.
  • Line 440. EC50 of PF74 can’t possibly be 1.239 nM. 1.239 uM sounds about right.
  • Lines 442-443. “there have been ongoing medicinal chemistry efforts to generate analogues less metabolically labile89”. Citation #89 described medicinal chemistry efforts generating novel PF74-like analogs, but no compounds were tested for metabolic stability. Instead, later citations #91-94 all addressed the metabolic stability issue via medicinal chemistry. The authors should consider reorganizing this part by combining all these citations. One possible way to do it is to cite #89, #91-94 after “low metabolic stability”, change the sentence to “there have been ongoing medicinal chemistry efforts to generate analogues with balanced potency and metabolic stability” (cite #91-94 herein), and delete the “Even with this success though, there remains ongoing medicinal chemistry work to develop alternative PF74 analogues with better metabolic stability and potency91–94”  (lines 459-460).
  • Lines 467-468. “as-yet-unclear mechanism: It’s reported that ebselen targets CA CTD by covalently crosslinking a cysteine residue. By the same token, the small and electrophilic ebselen is promiscuous by nature.

Reviewer 2 Report

In this review article, Wilbourne and Zhang detail many aspects of the interaction between antivirals (chemical and host-factors) and host-factors. They do a excellent job in detailing the structural knowledge that has been obtained over the years with different techniques and their context to infection, and their possible usage as antiviral targets for therapy. They do it very efficiently yet detailing them to a very specific level.

However, there have been some sections that I believe they could be slightly improved in the details, or expand to also discuss different hypotheses presented by many laboratories, as the authors seem to follow and cite only the articles that support the author’s views in some sections of the text.

Line 48- “and plays a role in viral DNA integration into the host genome”

Please consider modifying to: play a role in determining the integration sites of integration into the host genome;  to bring some non-ambiguous detail that recapitulates the studies that were cited.

Indeed, there as been a recent study (Burdick 2020) that suggested that the whole capsid is present at integration, with a new construct that insert a big protein into CA. However, there is a myriad of other studies that show heavy decrease of CA in the cytoplasm, and in some studies this has been associated with infection.

I am not aware of any study that relates capsid to integration itself, if it exists please cite. And the statement would be correct.

Line 49 “disrupting one or more of these processes has been shown to limit formation of new, infectious viral particles. “

This happens because the defect or abortion of infection happens in a step prior to integration. Please consider modifying to a clearer sentence, the way it was written it leads to the speculation that early steps of infection have  a mechanistic impact onto the late steps of infection.

Lines 61-63: “Some proposals suggested uncoating occurs in the cytoplasm shortly after fusion, but more popular hypotheses suggest uncoating takes place at the nuclear pore, or even that the capsid survives at least partially intact in the nucleus10–12. ”

I am not aware that science was a popularity contest, I heavily criticize the use of such terms and views.

To avoid speculation please define what you mean by uncoating, as all of the 3 studies that were cited support that CA is present and assembled into the viral complexes into the nuclear pore (or inside the nucleus). Did the authors mean that uncoating completes in a short period of time? Because none of the 3 cited studies argues so.
What these 3 studies argue differently is where the initiation of the CA shedding from the capsid occurs, while some argue that it initiates in the cytoplasm dependent of RT with CA retention until the nucleus(12), whether the capsid is completely intact at the nuclear pore (10), or whether the capsid is fully intact until integration (11). Some are subsets of data, some recapitulate the whole infectivity. But again, science can’t be a popularity contest and if there are different hypotheses, they should be discussed and the whole data presented in a comprehensive way.

There are other many studies that have studied these aspects showing progressive capsid shedding that extends from fusion to integration, others supporting uncoating only  at the NPC, only one so far fully supports an intact capsid at the integration site. The authors ignored decades of research into one sentence.

Here’s a short compilation of what the laboratories’ data have supported over time in peer-reviewer publications:

Early initiation of uncoating: Hope lab, Bishop Lab, Campbell lab, Diaz-Griffero Lab, Ambrose lab Pathak lab.

Cytoplasmic initiation and continutation in the nuclear envelope: Melykian Lab, Campbell Lab, Pathak lab.

Core intact until the NPC: Pathak Lab, Kraushlich lab, Campbell lab.

Completely happens at the integration site: Pathak lab.

CA detection in the nucleus: Kraushlich lab, Hope Lab, Di-Nunzio Lab, Melykian Lab, Manel lab.

Although it does not seem to be the main point of this review, I believe the authors should rephrase these statements to a more balanced stance, or describe exactly what all these studies showed. As one can easily gather, different articles from the same laboratory have supported different theories over the years, obtaining different results depending on the constructs used. Other aspect is that sometimes thousands of viral particles are analysed and only a subset of particles is used to support a theory, ignoring all the other viral behaviors present in the cell to only support one view.

78 CA is the sole constituent of the mature capsid, a conical structure with fullerene  geometry

Please consider changing to "fullerene-like" as it would be more correct. A fullerene is similar to a “round football” and not conical.

  1. Approximately 1500 CA monomers assemble into 200-250 hexamers and precisely 12 pentamers to make up the mature capsid4.

Have there been enough studies in present day to use the word “precisely” here? I'll leave it to the authors the prerogative to keep it as is.

90  and that the disturbing these results in the initiation of the uncoating process.

This was not show in any experimental infection study in cells or in vitro, it’s from a modeling study, and it has no connection HIV natural initiation of uncoating, but to a modeled defect in capsid integrity. The authors should rephrase this claim to reflect that.

92 - All-atom molecular dynamics simulations showed chloride ions and sodium  ions associated with capsid, providing one possible explanation for why CA monomers  assemble into tubes in vitro at high salt concentrations23.

Could the authors detail on the biochemical aspects of it? Basically, on how the high salt would favor tubes vs capsids. I believe it might facilitate the reading to a not biochemical specialized audience and even propel important studies.

Line 190 “. Christensen and colleagues demonstrated that reverse transcription takes place inside an intact or nearly intact capsid, using capsids derived from wild-type virions.”

Please rephrase or detail how much of reverse transcription was connected to which aspect of the capsid uncoating or state of the structure. The study showed a progressive shedding CA from the assemblies of capsid, then relating different stages of RT to the tomography images. Reverse transcription takes hours and not seconds, and it has been shown not to be a static binary system, but a plastic and interconnected system.

The fashion that the authors write does not fully recapitulate the data presented in the cited article. The same article also argues against an intact capsid at the nuclear pore as RT initiates in the cytoplasm as they show progressive CA shedding in a “cannibalized” capsid over time.

Line 242 CA (G89V),

please consider also adding and describing P90A, as the most recent studies have this mutation, as it has been shown to have less defects on infectivity comparsed to 89V, while still not interacting with CypA.

Line 275 When rhesus Trim5α binds the HIV capsid, it disrupts capsid stability, causing disassembly and initiating the innate immune response

I have the opinion that this claim is speculative/hypothesized. There have been studies, that the authors fail to cite and discuss, showing intact capsids trapped in proteasome incompetent TRIM bodies in cells, which are able to reverse transcribe where this dsDNA is cabable of integration in vitro. (Engelman, and Hope labs).

Indeed, naturally rhTRIM5 binding leads to destruction of the capsids before RT, but it has been shown to occur via the proteasome, not through the binding itself.

There have been studies that showed a TRIM lattice  surrounding intact capsids (in vitro and cryo-em recent studies). If the author’s claim was indeed that simple, it would have been very hard to show such lattice formation in intact cores as some articles have had (Pornillos, Jensen, and other labs).

I undertand that the authors have published data that supports this claim with in vitro, Cryo-EM, and biochemical studies( Zhao, 2011 Plos Pathogens) and it's a valid one indeed, however there is also data that supports other hypotheses and the authors should at least discuss such data and hypothesis.

  1. Nup153/Nup358

I could gather that the authors formed the text logic order to recapitulate the HIV life cycle from fusion to integration. Nup358 is located in the cytoplasmic side and Nup153 is associated with the nuclear side.

I suggest the authors to simply swap the order in this section of the textto keep the same text flow and logic. This, to first describe Nup358 and then Nup153.

336 - Nucleoporin 153 (Nup153) is required for HIV trafficking into the nucleus

The very nice studies from the Engelman lab that were cited, have defined a dependence to infection and binding to Nup153, not a nuclear translocation event itself (trafficking). There have been subsequent studies that have directly detailed aspects of nuclear import (Melikian lab and Pathak lab) that could be cited as well.

Typos:

Line 21 “diminishes it” (the viral infectivity)

Line 80. “due” is a word that emply negativity. Consider replacing by “This is thought to happen thanks to”.

Line 174, However should start the sentence not end it.

Reviewer 3 Report

The review article by Wilbourne and Zhang, discusses the several efforts put into understanding the structural biology of the HIV-1 capsid protein, which is a viable target for drug development. The paper is overall well written and discusses appropriately the limitations and key differences between the structures of capsid obtained through various techniques (CryoEM/cryoET, NMR, and X-ray crystallography). This is an important and timely review of the literature. However, I found a series of absent references, typographical and conceptual errors in the review, pointed out below:

Minor Errors

Line 16 : “to close to the cone”, this should read “to close the cone”

Lines 62-64: “ Some proposals suggested uncoating occurs in the cytoplasm shortly after fusion, 62 but more popular hypotheses suggest uncoating takes place at the nuclear pore, or even 63 that the capsid survives at least partially intact in the nucleus10–12”  Several references are missing.

Line 73: CA is a 24 kDa protein (not 25)

Lines 81-82: somewhat repetitive to abstract.

Lines 96-97: “channels through which nucleotides could transit to facilitate re-96 verse transcription in vivo”  is missing a reference.

Lines 143-146: Can the authors describe the differences found between NMR and CryoEM maps?

Lines 242: “reduces viral infectivity” should be changed to “modulates viral infectivity” for appropriateness

Conceptual discussions

Lines 186-187: “which were observed in CA tubes of different the helical symmetry.” Should read “different helical symmetry”.

The paper described in Lines 189-194 investigated in vitro reverse transcription and observed capsid defects. The notion that RT stimulates uncoating was originally described by others using core-pelleting assays, CsA washout assay and imaging, these papers are not cited. The data present by Christensen argue that in the presence of IP6, Mellitin, dNTPs and rNTPs conical or partial cores are recovered at time points when full length DNA is detected, as such the discussion should explicitly state that this is an invitro system.

Lines 258-259: “This multi-valent binding of CypA to the assembled capsid explains its stabilizing effect on the viral capsid.” I suppose the multivalency proposed here indicates to CypA binding to multiple sites. Perhaps Multi-modal binding is a more appropriate term. I say this because multivalency binding will mean CypA is a dimer or tetramer and also high-avid interactions between CypA-CA, however several prior papers have shown that CypA-CA interactions is at best 1-20uM.

Lines 436-439: “Recent studies using PF74 have demonstrated that it inhibits integration of viral DNA into the host genome, demonstrating a role for CA in this process and lending credence to hypotheses that uncoating is not complete prior to nuclear entry”

This is a highly contentious notion, especially because CA has no role in integration, but has a role in integration targeting into genes. Other studies referenced in this review have suggested otherwise, that the inhibition of integration is due to the loss of HIV-1 speckle contacts due to CPSF6 washout in the same speckles, and not due to inducing CA-disassembly. In fact as shown in papers  from multiple labs, one needs much higher concentration of PF74 (10-25 uM) to inhibit integration than the IC50 (0.5 uM), any comments on this? The effect of PF74 should be discussed more appropriately.
